# Predictive Factors of HIV-1 Drug Resistance and Its Distribution among Female Sex Workers in the Democratic Republic of the Congo (DRC)

**DOI:** 10.3390/ijerph19042021

**Published:** 2022-02-11

**Authors:** Godefroid Mulakilwa Ali Musema, Pierre Zalagile Akilimali, Takaisi Kikuni Ntonbo za Balega, Désiré Tshala-Katumbay, Paul-Samson Dikasa Lusamba

**Affiliations:** 1Kinshasa School of Public Health, University of Kinshasa, Kinshasa P.O. Box 11850, Democratic Republic of the Congo; gmusema2014@gmail.com (G.M.A.M.); luspauls@gmail.com (P.-S.D.L.); 2Pharmaceutical Sciences Faculty, University of Kinshasa, Kinshasa P.O. Box 11850, Democratic Republic of the Congo; takaisik@unikin.ac.cd; 3Department of Neurology, School of Medicine and School of Public Health, Oregon Health & Science University, Portland, OR 97239, USA; tshalad@ohsu.edu; 4Department of Neurology, School of Medicine, University of Kinshasa, Kinshasa P.O. Box 11850, Democratic Republic of the Congo; 5Institut National de Recherches Biomédicales, Kinshasa P.O. Box 11850, Democratic Republic of the Congo

**Keywords:** HIV-1 drug resistance, HIV-1 drug resistance mutations, HIV-1 viral load, HIV-1–syphilis coinfection, female sex workers, Democratic Republic of the Congo

## Abstract

The predictive factors of HIV-1 drug resistance and its distribution are poorly documented in female sex workers (FSWs) in the Democratic Republic of the Congo (DRC). However, the identification of predictive factors can lead to the development of improved and effective antiretroviral therapy (ART). The objective of the current study was to determine the predictive factors of HIV-1 drug resistance and its distribution based on FSWs in the studied regions in the Democratic Republic of the Congo (DRC). HIV-positive FSWs who were diagnosed as part of the DRC Integrated Biological and Behavioral Surveillance Survey (IBBS) were included in this study. A total of 325 FSWs participated. The HIV-1 viral load (VL) was measured according to the Abbott m2000sp and m2000rt protocols. The homogeneity chi-square test was conducted to determine the homogeneity of HIV-1 drug resistance distribution. Using a significance level of 0.05, multivariate analyses were performed to identify factors associated with HIV-1 drug resistance to ART. HIV drug resistance mutation (HIVDRM) distribution was homogeneous in the three study regions (*p* = 0.554) but differed based on the HIV-1 VLs of the FSWs. FSWs with high HIV-1 VLs harbored more HIVDRMs (*p* = 0.028) of predominantly pure HIV-1 strains compared with those that had low HIV-1 VLs. Sexually transmitted infection (STI) history (aOR [95%CI] = 8.51 [1.62, 44.74]), high HIV-1 VLs (aOR [95%CI] = 5.39 [1.09, 26.74]), and HIV-1–syphilis coinfection (aOR [95%CI] = 9.71 [1.84, 51.27]) were associated with HIV drug resistance among FSWs in the DRC. A history of STIs (e.g., abnormal fluid) in the 12 months prior to the survey, a high HIV-1 VL, and HIV-1–syphilis coinfection were associated with HIV-1 drug resistance among FSWs in the DRC. Efforts should be made to systematically test for other infections which increase the HIV-1 VL, in the case of HIV-1 coinfection, in order to maintain ART effectiveness across the DRC.

## 1. Introduction

### 1.1. Background of the Study

Despite its declining global prevalence, human immunodeficiency virus (HIV) infection remains a public health concern worldwide due to the morbidity and the mortality it causes in affected populations. Those most at risk of HIV infection include female sex workers (FSWs), men who have sex with men (MSM), and people who inject drugs (PWID), especially in sub-Saharan Africa (SSA) [1,2,3,4].

FSWs continue to bear the heavy burden of HIV risk and infection despite the many efforts made toward prevention and treatment in SSA [4]. According to the Joint United Nations Programme on HIV/AIDS (UNAIDS) 2017 data, approximately 64% of the 5000 new HIV infections that occurred globally during that year were in SSA [5].

Although large-scale antiretroviral therapy (ART) has been beneficial in treating HIV patients due to its impact in reducing mortality and morbidity, it has also prompted the rapid emergence of HIV drug resistance (HIVDR) around the world, especially in SSA [6,7].

In the Democratic Republic of the Congo (DRC), HIV prevalence among FSWs was 14 times higher in 2005–2006 [8] and 6 times higher in 2012–2013 [9], compared with the global DRC prevalence rates. The poverty, as well as the lack of familial, social, financial, and legal support systems that are characteristic of life in SSA, particularly the DRC, have directed FSWs into this high-risk occupation that requires them to have multiple, often unknown sex partners. As a result, this can lead to multiple HIV infections and superinfections. These multiple infections and superinfections that characterize FSWs and their typically male clientele can promote the transmission of HIVDR [10,11,12,13], increase the HIV viral load (VL), and cause rapid emergence of HIVDR within the general population [14,15].

Overall, HIV-1 recombinant forms dominate the HIV-1 molecular epidemiology in the DRC. This situation is the result of multiple HIV-1 infections and superinfections often found in FSWs [16,17]. Several studies have shown that FSWs and their male clientele play an important role in the evolution of the HIV epidemic and in HIV infection transmission in the general population [18,19,20]. Increasing cases of HIVDR to ART have been observed in countries such as Uganda [6] and South Africa [21].

In SSA, the high HIV VL in people living with HIV is a result of the high rates of HIV coinfection with other infections. Indeed, HIV coinfection with other infections (including sexually transmitted infections (STIs), tuberculosis, and hepatitis) aggravates HIV infection, increases its VL and facilitates its transmission [22,23]. Musema et al. reported a strong association between coinfection of HIV-1 and high HIV-1 VL in FSWs from the DRC [24].

In addition, an increase in HIV drug resistance levels between 2004 and 2014 in low-and middle-income countries was reported by the World Health Organization (WHO, Geneva, Switzerland) [25]. This increase concerned non-nucleoside reverse transcriptase inhibitors (NNRTIs) and was mostly observed in individuals who were naïve to ART [26]. NNRTIs are the class of inhibitors used in several first-line antiretroviral therapy regimens in those countries. However, in addition to the large-scale availability of ART worldwide, particularly in SSA, regular virological monitoring (especially HIV-1 VL and genotypic tests for drug resistance) must accompany these efforts to prevent the emergence of drug resistance.

Several factors contribute to the increased risk of acquiring HIVDRMs which are resistant to ART; therefore, it is crucial to identify these predictive factors of HIV-1 drug resistance in FSWs from the DRC, which are conduits for HIV transmission to the local population.

### 1.2. Objectives of the Study

The overall objective of this study is to determine the predictive factors of HIV-1 drug resistance and its distribution according to the study groups in the DRC.

### 1.3. Specific Aims and Hypothesis

The specific objectives of this study are:To determine the sociodemographic characteristics of the HIV-positive FSWs in the DRC;To determine the prevalence of HIV drug resistance mutations (HIVDRMs) based on the HIV-1 VL in FSWs in the DRC;To determine the HIVDRM distribution based on the study groups in the DRC;To identify the factors associated with HIVDRMs in FSWs in the DRC.

## 2. Methods

### 2.1. Data Source

Data of FSWs from the DRC HIV/STI Integrated Biological and Behavioral Surveillance Survey (IBBS), collected between December 2012 and January 2013 (DRC IBSS-2012), [9] were used in this study. Sociodemographic data, biological samples (DBS and plasma), and data on HIV risk behavior were collected and analyzed.

### 2.2. IBBS Design, Setting and Population

The purpose of the DRC IBSS-2012 was to assess behavioral changes in relation to HIV and syphilis, in addition to the prevalence of HIV risk behaviors among four high-risk populations, namely “street children” (i.e., homeless children), truck drivers, miners, and FSWs. Only FSWs with HIV-positive test results were chosen as subjects for this study.

Since FSWs are often a transient population, time–location sampling (TLS) was used to recruit study subjects aged between 15 and 49 years, corresponding to the reproductive age. Each study participant consented, and confidentiality was guaranteed. No reward was offered for participation in the study. Study participants between the ages of 15 and 17 years were considered emancipated minors and capable of making their own decisions.

Biological samples (e.g., plasma and dried blood spot (DBS)) were taken after a second consent was given by participants, and the samples were tested in the field for HIV and syphilis. All samples were labeled with a ten-digit study number not linked to the individual’s name. HIV and syphilis diagnoses were carried out using rapid tests, according to the DRC nationally approved protocols at the time of the survey. Positive tests were confirmed using enzyme-linked immunosorbent assay (ELISA) at the National HIV/AIDS Reference Laboratory.

The survey was carried out from December 2012 to January 2013 in 11 cities in former provinces. These cities were divided into three pools: (1) West region (Kinshasa, Matadi, Mbandaka, and Kikwit); (2) Northeast region (Kisangani, Kindu, Goma, and Bukavu); and (3) South region (Lubumbashi, Kananga, and Mbuji-Mayi). A total of 325 FSWs were included in this study, all of whom were HIV-positive, and whose DBS and plasma samples could be acquired.

### 2.3. Sociodemographic Characteristics and HIV Risk Behavior Data

Sociodemographic data and HIV risk behaviors were analyzed using SPSS version 26.0 and Stata version 14.0. Age was divided into three categories (˂20, 20–24, and 25–49 years old) that were in line with epidemiological FSW classifications. Education levels were classified as no formal education, primary and secondary education, and higher education. Duration as a sex worker was categorized as ˂5 or ≥5 years. Two STI histories were identified in terms of the presence of genital ulcerations (yes or no) and abnormal genital secretions (yes or no) within the 12 months prior to the survey. Alcohol consumption (yes or no) and marital status (unmarried or married) were also recorded. HIV-1 VL was grouped into two categories according to the WHO classification (i.e., <3.0 log10 HIV RNA/mL or ≥3.0 log10 HIV RNA/mL) and the HIV-1–syphilis coinfection status (yes or no).

### 2.4. Laboratory Procedures

#### 2.4.1. Measuring HIV-1 Viral Load

Plasmatic HIV-1 VL was measured at the National HIV/AIDS Reference Laboratory, according to the Abbott M2000RT assay protocol and then expressed in log10 HIV RNA/mL. HIV-1 VL was categorized into high VL (≥3.0 log10 HIV RNA/mL) and low VL (˂3.0 log10 HIV RNA/mL) according to WHO directives on HIV treatment and prevention [27].

#### 2.4.2. HIV-1 Subtype Identification and Detection of Drug Resistance Mutations

The determination of the HIV-1 group M subtypes from pol- and env-gene sequences, as well as the detection of drug resistance mutations, were performed at the Center for Virology at the University of Nebraska–Lincoln (UNL), USA [17].

#### 2.4.3. Syphilis Testing

Syphilis status was determined on site using both the Determine^®^ Syphilis TP test (Inverness Medical Professional Diagnostics, Princeton, NJ, USA) and the rapid plasma reagin (RPR) carbon test (Cypress Diagnostics, Langdorp, Belgium). Determine Syphilis TP (100.0% sensitivity) was used as a screening test. All specimens positive with the Determine Syphilis TP test were subsequently tested with the RPR test for confirmation. Those reactive to RPR were reported as syphilis-positive [17].

### 2.5. Study Variables

For this study, we identified the variables of interest, including HIV-1 drug resistance as a dependent variable. We identified the following as independent variables: age, education level, duration in sex profession, STI history (e.g., presence of genital ulcerations and abnormal genital secretions in the 12 months prior to the survey), alcohol use, marital status, and HIV-1 VL.

### 2.6. Statistical Analysis

SPSS v. 26.0 and Stata version 14.0 were used for various statistical tests and analyses, including one-way ANOVA and the multivariate logistic regression model. A reading of *p* ≤ 0.05 was considered to indicate statistically significant differences. Continuous variables were summarized by means and standard deviation (SD). The homogeneity chi-square test was performed to test the HIV-1 drug resistance distribution homogeneity. For viral load, some data are missing due to poor sample storage and insufficient quantity of samples during quality control at the National HIV/STI Reference Laboratory in Kinshasa. For pharmaco-resistance, there were only 93 dry blood spot (DBS) samples, for which sequencing for the determination of genetic mutations of resistance was successfully performed, due to the conditions and duration of storage of DBS samples. Logistic regression models were fitted to identify factors associated with HIV-1 drug resistance and *p* ˂ 0.05 was considered to indicate statistically significant differences.

### 2.7. Ethical Approval

The original DRC HIV/STI IBBS-2012 protocol was approved by the Tulane Human Research Protection Program Biomedical and Social Behavioral Institutional Review Boards FWA000002055 (IRB Reference#: 12-347922). Ethical clearance was also obtained from the Kinshasa School of Public Health (KSPH) IRB. This study was conducted according to the guidelines of the Declaration of Helsinki and received IRB approval from Johns Hopkins University (IRB No.: 00009677) and the Kinshasa School of Public Health (ESP/CEI/030B/2019).

## 3. Results

### 3.1. Sociodemographic Characteristics and HIV Risk Behaviors of FSWs

Table 1 summarizes the characteristics of the FSWs diagnosed as HIV-positive, with a mean age of 30.5 (95%CI: 29.66 to 31.35) years. More than three-quarters (79.0%) were aged between 25 and 49 years. Roughly half (47.5%) had reached secondary school and above while 15.7% had never attended school. More than half (59.0%) of the HIV-positive FSWs reported having at least five years of experience in the trade. More than three-quarters (77.3%) of HIV-positive FSWs declared having consumed alcohol, and more than half (66.7%) admitted to having been married before. Regarding their STI histories, 23.6% and 30.0% admitted to having genital ulcerations and abnormal genital secretions, respectively, within the 12 months prior to the survey. Of the 265 HIV-positive FSWs in whom the VL was successfully measured, 46.4% had a high HIV-1 VL (≥3.0 log10 HIV RNA/mL). Among the HIV-positive FSWs from the DRC, 75 (23.1%) were coinfected with HIV-1 and syphilis. Finally, of the 93 HIV-positive FSWs on whom the HIVDR test was successfully performed, 21.5% harbored HIVDRMs.

### 3.2. HIV-1 Drug Resistance Distribution

HIV-1 drug resistance distribution was homogeneous across the three study regions (*p* = 0.554). However, the highest resistance rate was observed in the Northeast region (10/31 (32.2%)), followed by the West region (3/16 (23.1%)) and the South region (7/37 (18.9%)).

### 3.3. HIV-1 Drug Resistance Mutations According to HIV-1 VL

Table 2 shows the HIVDRMs and HIV-1 VL scores. The FSWs with a high HIV-1 VL (32.6%) harbored significantly more HIVDRMs compared with those that had a low HIV-1 VL (11.4%).

Table 2 shows the predominance of M184V/I (80.0%) and K103N (38.5%) for NRTI and NNRTI, respectively.

### 3.4. Factors Associated with ART Resistance in FSWs

Table 3 shows that drug resistance to ART was associated with STI history and abnormal genital secretions in the 12 months prior to the survey (adjusted odds ratio (aOR) [95%CI] = 8.51 [1.62, 44.74]), as well as with high HIV-1 VL (aOR [95%CI] = 5.39 [1.09, 26.74]) and HIV-1–syphilis coinfection (aOR [95%CI] = 9.71 [1.84, 51.27]). In fact, HIV-1-positive FSWs who had a history of STIs were 8.51 times more likely to have ART resistance compared with those without an STI history (*p* = 0.011). Similarly, FSWs with a high HIV-1 VL were 5.39 times more likely to show HIV-1 drug resistance than those with a low HIV-1 VL (*p* = 0.039). Finally, HIV-positive FSWs coinfected with syphilis had a 9.71 times higher risk of HIV-1 drug resistance compared with those who were not coinfected (*p* = 0.007).

## 4. Discussion

The objective of this study was to determine the predictive factors of HIV-1 drug resistance and its distribution in the DRC based on HIV-positive FSWs in the DRC. The mean age of the participants in this study was 30.5 years. The HIV-1 drug resistance distribution was homogeneous across the three study regions. The presence of abnormal genital secretions, a high HIV-1 VL, and HIV-1–syphilis coinfection were associated with the HIV-1 drug resistance in FSWs in the DRC.

### 4.1. Sociodemographic Characteristics and HIV Risk Behaviors of FSWs

The mean age of FSWs in the DRC (30.5 years) was slightly lower than in Uganda (32.5) [28] and was similar to Andhra Pradesh in India (30.5 years) [29].

### 4.2. HIV-1 Drug Resistance Mutations According to HIV-1 VL

Overall, the vast majority of HIV-positive FSWs harboring HIVDRM had high HIV-1 VLs, (*p* = 0.028), as shown in Table 2. These findings suggest that FSWs with high plasma HIV-1 VLs are more likely to have HIVDRMs. High HIV-1 VLs have been attributed to multiple infections or superinfections that are often found in FSWs [30]. They may also be due to HIV-1–syphilis coinfection [24] or with other STIs, such as gonorrhea, chlamydia, herpes, etc. [31]; these were not considered in this study. The potential risk of acquisition and transmission of HIVDRMs in FSWs has the potential to increase the HIV risk for the general population as well as for their clients, and can likely be attributed to the realities of sex work, such as having multiple sexual partners and the inconsistent use of prophylactics.

Moreover, we found that among FSWs with high HIV-1 VLs, the most predominant HIVDRMs observed in this study were M184V/I and K103N for NRTIs and NNRTIs, respectively. These results are aligned with the findings of a Ugandan study of HIV-positive FSWs [25] and a study of the general Rwandan population [32]. HIVDRMs confer major resistance to lamivudine (3TC), tenofovir (TDF), nevirapine (NVP), and efavirenz (EFV), which are part of a therapeutic combination of first- and second-line treatments widely used in ART in numerous resource-limited regions [25]. This considerably impacts the effectiveness of first- and second-line ART in the DRC, which are typically TDF/AZT + 3TC + DTG/EFV and AZT/TDF + 3TC + DTG/LPV, respectively. These findings underscore the importance of including proactive testing for HIV-1 VLs and HIVDRMs in the services offered to FSWs. Our findings suggest the possibility that one-fifth of FSWs have a virus that is resistant to treatment with drugs of first-line ART.

Despite the lack of information regarding the treatment of HIV-1-positive FSWs who participated in this study, our findings and those of a previous study [17] confirm that FSWs in the DRC are at risk of acquiring HIVDRMs via sexual transmission. Our findings also underscore the importance of organizing and decentralizing care services for FSWs across provincial capitals so that FSWs have access to effective healthcare and medical interventions. These services need to proactively test and analyze the HIV-1 strains and HIVDRMs in FSWs as well as evaluate their HIV-1 VL. Such procedures and the acquired information will support rapid, effective decision making in the event of therapeutic failures.

### 4.3. Factors Associated with ART Resistance in FSWs in DRC

STI history in the 12 months prior to the survey, high HIV-1 VL, and coinfection with syphilis were factors that we found to be associated with HIV-1 drug resistance in HIV-positive FSWs. Indeed, the interaction between STIs and HIV infection promotes viral multiplication by attracting immune cells to the infection site so that they can be infected by the viruses as well as by altering the mucosal barrier. This exponential viral multiplication can lead to an increase in viral load with a high probability of creating genetically resistant mutations [33,34,35]. However, our findings diverged from many studies that have established a significant association between HIV-1 drug resistance and younger people living with HIV [31,35].

### 4.4. Limitations

This study had some limitations, such as a lack of information on the treatment status of participants which would allow the identification of FSWs in cases of treatment failure. While the HIV epidemic is raging across the DRC in urban and rural areas alike, it was not possible to collect data from rural areas in this study. Since the interaction of HIV with syphilis is very complex with mutual negative impacts, we could not determine which of the two had first infected the FSWs. In addition, the fact that data were collected ten years ago is another limit. Finally, although we have used multivariate regression, we cannot exclude the potential confounding effect of the other variables such as genital excrescence.

## 5. Conclusions and Implications for Translation

In this study, we showed that HIV-positive FSWs in the DRC who had high HIV-1 VLs harbored more HIVDRMs, and HIV-1 recombinant forms were less affected by NRTIs. HIV-1–syphilis coinfection, a high HIV-1 VL, and a history of STIs (e.g., abnormal genital secretions in the 12 months prior to the survey) were associated with HIVDRMs among FSWs. To improve ART effectiveness among high-risk HIV populations such as FSWs, efforts should be made to search for HIV-1 coinfection with other infections that may increase the HIV-1 VL. It is also critical to assess patients for HIVDRMs before initiating any treatment of HIV infections.

## Figures and Tables

**Table 1 ijerph-19-02021-t001:** Sociodemographic characteristics and risk behaviors of HIV-positive FSWs from the DRC.

Characteristics	*n*	%
Age category (years) (*n* = 324)		
˂20	21	6.5
20–24	47	14.5
25–49	256	79.0
Education level (*n* = 324)		
No formal education	51	15.7
Primary	119	36.7
Secondary and above	154	47.5
Duration in sex trade (years) (*n* = 323) *		
≥5	131	40.6
˂5	192	59.4
Alcohol use (*n* = 322) *		
Yes	249	77.3
No	73	22.7
Marital status (*n* = 324)		
Married	216	66.7
Unmarried	108	33.3
Genital ulcerations in 12 months prior to survey (*n* = 322) *		
Yes	76	23.6
No	246	76.4
Abnormal genital secretions in 12 months prior to survey(*n* = 323) *		
Yes	97	30.0
No	226	70.0
HIV-1–syphilis coinfection status (*n* = 324)		
Yes	75	30.0
No	249	76.9
HIV-1 viral load (log_10_ HIV RNA/mL) (*n* = 265)		
≥3.0	123	46.4
˂3.0	142	53.6
HIV drug resistance (*n* = 93) **		
Yes	20	21.5
No	73	78.5

*: There were 1, 2, 3, or 12 missing values. **: HIV drug resistance was assessed only for 93 patients on whom the HIVDR test was successfully performed. DRC: Democratic Republic of the Congo. FSW: female sex worker.

**Table 2 ijerph-19-02021-t002:** Summary of HIV-1 drug resistance mutations observed in 20 FSWs in the DRC according to HIV-1 VL.

HIV-1 VL	DR Prevalence	ID	HIV Clade	HIV Drug Mutations (Scores)			*p*
			POL	PI	NRTI	NNRTI	
˂3.0 log10 RNA copies/mL	4/35 (11.4%)	4-B7	D			E138G (45)	0.028
		2-A7	C		K65R (150)		
		2-D4	A1/C	L33F (15)			
		1-E8	A1/G/A1/F1/C/G/K		M184V (115)	G190A (130)	
≥3.0 log10 RNA copies/mL	14/43 (32.6%)	3-A3	D		K219R (20)		
		2-A6	A			K103N (120)	
		4-G5	G			V179E (40)	
		3-F6	C		M184V (115)	A98G/V179D/Y181C (295)	
		2-C3	A	L33F (15)			
		2-E2	CRF05_DF			K103N (120)	
		2-E8	A		M184V (115)	A98G/K103N (190)	
		2-C6	C			E138A (25)	
		2-H1	C		M184I (115)		
		3-D4	A1/K/A1/C			V179T (0)	
		3-I7	A		M184V (115)	K103N (120)	
		3-E6	B/C			E138A (25)	
		2-D8	D/G/H/A1			K103N (120)	
		1-B7	A	I54V (30)			

DR: Drug resistance; ID: identification; VL: Viral load; NRTI: Nucleoside/nucleotide reverse transcriptase inhibitors; NNRTI: Non-nucleoside reverse transcriptase inhibitors; PI: protease inhibitors.

**Table 3 ijerph-19-02021-t003:** Factors associated with resistance to ART among HIV-seropositive FSWs from DRC.

Characteristics	Resistance to ART		Crude OR(95% CI)	*p*	Adjusted OR(95% CI)	*p*
	Yes *n* (%)	No *n* (%)				
Age category (years)						
- <20	3 (33.3)	6 (66.7)	1		1	
- 20–24	4 (26.7)	11 (73.3)	0.73 (0.12–4.39)	0.728	0.30 (0.01–6.03)	0.428
- 25–49	13 (18.8)	56 (81.2)	0.46 (0.10–2.11)	0.320	0.74 (0.06–8.90)	0.812
Education level						
- No education	4 (23.5)	13 (76.5)	1		1	
- Primary	6 (17.6)	28 (82.4)	0.70 (0.17–2.90)	0.619	0.83 (0.13–5.30)	0.840
- Secondary	10 (23.8)	32 (76.2)	1.02 (0.27–3.83)	0.982	2.48 (0.38–16.11)	0.341
Duration in sex trade (years)						
- ˂5	12 (20.3)	47 (79.7)	1		1	
- ≥5	8 (23.5)	26(76.5)	1.21 (0.44–3.33)	0.719	1.65 (0.28–9.64)	0.576
Alcohol use						
- No	59 (80.8)	14 (19.2)	1		1	
- Yes	15 (75.0)	5 (25.0)	1.51 (0.47–4.91)	0.491	2.18 (0.37–12.75)	0.386
Marital Status						
- Unmarried	47 (64.4)	26 (35.6)	1		1	
- Married	10 (50.0)	10 (50.0)	1.81 (0.67–4.91)	0.245	3.10 (0.52–18.48)	0.215
Genital ulcerations in 12 months prior to survey						
- No	11 (15.9)	58 (84.1)	1		1	
- Yes	9 (37.5)	15 (62.5)	3.16 (1.11–9.02)	0.031	1.13 (0.16–7.84)	0.901
Abnormal genital secretions in 12 months prior to survey						
- No	8 (12.5)	56 (87.5)	1		1	
- Yes	12 (41.4)	17 (58.6)	4.94 (1.74–14.07)	0.003	8.51 (1.62–44.74)	0.011
HIV-1 Viral Load						
- <3.0 log 10 HIV RNA/mL	4 (11.4)	31 (88.6)	1		1	
- ≥3.0 log 10 HIV RNA/mL	14 (32.6)	29 (67.4)	3.74 (1.10–12.69)	0.034	5.39 (1.09–26.74)	0.039
HIV-1–syphilis coinfection status						
- No	8(12.7)	55(87.3)	1		1	
- Yes	12(40.0)	18(60.0)	4.58 (1.62–12.98)	0.004	9.71 (1.84–51.27)	0.007

Mean VIF = 2.22. ART: Antiretroviral Therapy; OR: odds ratio; CI: confidence interval; RNA: ribonucleic acid.

## Data Availability

Data are available by request at gmusema2014@gmail.com (G.M.A.M).

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
