# Peer review of "Predictive Factors of HIV-1 Drug Resistance and Its Distribution among Female Sex Workers in the Democratic Republic of the Congo (DRC)"

_ijerph, 2022, doi:10.3390/ijerph19042021_

Round 1

Reviewer 1 Report

In this manuscript, the authors investigated the association of HIV-1 drug resistance and age, education, duration in sex work, alcohol use, marital status, STI, HIV-1–syphilis coinfection status, and HIV-1 viral load of female sex workers. They concluded that HIV-1–syphilis coinfection, a high HIV-1 VL, and a history of STIs were associated with HIV-1 drug resistance among female sex workers. The factor that associates with HIV-1 drug resistance may be a good indicator for drug resistance of HIV-1 patients and thus to guide the therapy.

This manuscript is well organized. The results are clearly described. The references are correctly cited. The limitations are well discussed. I just have some suggestions to further improve the solidarity of this manuscript.

  1. In line 79-80, the statement “This increase concerned non-nucleoside reverse transcriptase inhibitors (NNRTIs) and was mostly observed in individuals who were naïve to ART” need a reference.
  2. In line 131, age was divided into three categories. This divide seems meaningless to me. Is there any difference, in terms of healthy condition, HIV-1 infection rate, mortality rate, or other factors that may associate with drug resistance, between 20-24 group and 25-49?
  3. In line 135, genital excrescence is also a significant symptom for many STIs. Why do the authors not include this?
  4. In line 181 and 184, the reason that data of HIV viral load and drug resistance of many subjects are missing should be provided here.
  5. The syphilis infection status was acquired by questionnaire in this study. However, the syphilis has latent stage, questionnaire is not a reliable way. Since the researchers have the blood sample of the subjects, treponemal antibody tests will be a good choice for confirming the syphilis infection.
  6. In line 234-236, the age seems similar between the DRC (30.5 years) and Tanzania and the Dominican Republic (31.7 years). Therefore, it cannot be concluded that “This is attributed to young women in the DRC engaging in sex work at an earlier age due to the aforementioned lack of social and societal support systems”.

There are also some minors suggestions:

  1. In line 191-124, this paragraph is kind of the same as the 3.4 section.
  2. The abbreviation aOR has not been explained.

Author Response

In this manuscript, the authors investigated the association of HIV-1 drug resistance and age, education, duration in sex work, alcohol use, marital status, STI, HIV-1–syphilis coinfection status, and HIV-1 viral load of female sex workers. They concluded that HIV-1–syphilis coinfection, a high HIV-1 VL, and a history of STIs were associated with HIV-1 drug resistance among female sex workers. The factor that associates with HIV-1 drug resistance may be a good indicator for drug resistance of HIV-1 patients and thus to guide the therapy.

This manuscript is well organized. The results are clearly described. The references are correctly cited. The limitations are well discussed. I just have some suggestions to further improve the solidarity of this manuscript.

 Authors’ response: Thanks

  1. In line 79-80, the statement “This increase concerned non-nucleoside reverse transcriptase inhibitors (NNRTIs) and was mostly observed in individuals who were naïve to ART” need a reference.

 Authors’ response: We have added the reference

  1. In line 131, age was divided into three categories. This divide seems meaningless to me. Is there any difference, in terms of healthy condition, HIV-1 infection rate, mortality rate, or other factors that may associate with drug resistance, between 20-24 group and 25-49?

 Authors’ response: These categories are used in FSW classifications

  1. In line 135, genital excrescence is also a significant symptom for many STIs. Why do the authors not include this?

 Authors’ response: Not included because it was not mentioned in the survey questionnaire but we have mentioned as limit.

  1. In line 181 and 184, the reason that data of HIV viral load and drug resistance of many subjects are missing should be provided here.

 Authors’ response: We have added and you can read in the method section: For viral load, some data are missing due to poor sample storage and insufficient quantity of samples during quality control at the National HIV/STI Reference Laboratory in Kinshasa. For Pharmaco-resistance, there were only 93 dry blood spot (DBS) samples for which sequencing for the determination of genetic mutations of resistance was successfully performed due to the conditions and duration of storage of DBS samples.

  1. The syphilis infection status was acquired by questionnaire in this study. However, the syphilis has latent stage, questionnaire is not a reliable way. Since the researchers have the blood sample of the subjects, treponemal antibody tests will be a good choice for confirming the syphilis infection.

Authors’ response: we have used a biological test and you can read it in the new version as follow: «Syphilis status was determined on site using both the Determine® Syphilis TP test (Inverness Medical Professional Diagnostics, Princeton, NJ) and the rapid plasma reagin (RPR) carbon test (Cypress Diagnostics, Langdorp, Belgium). Determine Syphilis TP (100.0%, sensitivity) was used as a screening test. All specimens positive with the Determine syphilis TP test were subsequently tested with the RPR test for confirmation. Those reactive to RPR were reported as syphilis positive [17]. »

  1. In line 234-236, the age seems similar between the DRC (30.5 years) and Tanzania and the Dominican Republic (31.7 years). Therefore, it cannot be concluded that “This is attributed to young women in the DRC engaging in sex work at an earlier age due to the aforementioned lack of social and societal support systems”.

Authors’response: We have deletetd this in the current version

There are also some minors suggestions:

  1. In line 191-124, this paragraph is kind of the same as the 3.4 section.

Authors’response: We have deletetd o,ne of the statement and keep only one because of the duplicated statement,  in the current version

  1. The abbreviation aOR has not been explained.

Authors’response: adjusted Odds ratio=aOR and we have edited accordingly.

Reviewer 2 Report

In this study the authors aimed at search for factors associated with HIV-1 drug resistance (HIV DR) to antiretroviral treatment (ART) in female sex workers (FSWs) in the Democratic Republic of the Congo (DRC) as well as for the HIV DR distribution in three DRC areas. The data were collected ten years ago.

The study included 325 patients with informed consent. Sociodemographic characteristics and HIV risk behaviors were carefully analyzed in all participants.

A standard set of experimental and statistical methods was used in the study.

In 265 HIV-positive FSWs viral load was successfully measured, the sequencing was successful in 93 samples of blood cells or plasma. HIV-1 drug resistance mutations (DRMs) were observed in 20 FSWs (21.5%). The most prevalent mutations were M184V and K103N which corresponds to the ART regimens used in DRC.

According to the reasonable conclusions of the authors, HIV-1–syphilis coinfection, a high HIV-1 viral load, and a history of sexually transmitted infections were associated with HIV DRMs. HIV-1 drug resistance distribution was homogeneous across the three study regions.

Other findings are less statistically sound and need to be corrected or repealed; for example, the phrases “This abovementioned table also indicated the predominance of pure HIV-1 subtype  (71.4%, of which A = 50.0%, C = 30.0%, D = 10.0%, G = 10.0%) compared with HIV-1 recombinant forms in FSWs that had a high HIV-1 VL, while this HIV-1 strain was equally 209 distributed (50.0%) in FSWs that had a low HIV-1 VL score” and “HIV-1 recombinant forms were less affected by 288 NRTIs” clearly need additional statistical support.

The main limitation of the work, as noted by the authors, is the lack of information about the treatment of patients. This makes it impossible to draw conclusions about the prevalence of mutations before treatment start. In this regard, the statement "around one-fifth of FSWs could fail to respond to first-line ART due to HIVDRMs" needs to be corrected. It would be more correct to say that one-fifth of FSWs have a virus that is resistant to treatment with drugs of first-line ART.

There are a few typos in the text.

The article, in our opinion, needs correction and subsequent publication.

Author Response

In this study the authors aimed at search for factors associated with HIV-1 drug resistance (HIV DR) to antiretroviral treatment (ART) in female sex workers (FSWs) in the Democratic Republic of the Congo (DRC) as well as for the HIV DR distribution in three DRC areas. The data were collected ten years ago.

The study included 325 patients with informed consent. Sociodemographic characteristics and HIV risk behaviors were carefully analyzed in all participants.

A standard set of experimental and statistical methods was used in the study.

In 265 HIV-positive FSWs viral load was successfully measured, the sequencing was successful in 93 samples of blood cells or plasma. HIV-1 drug resistance mutations (DRMs) were observed in 20 FSWs (21.5%). The most prevalent mutations were M184V and K103N which corresponds to the ART regimens used in DRC.

According to the reasonable conclusions of the authors, HIV-1–syphilis coinfection, a high HIV-1 viral load, and a history of sexually transmitted infections were associated with HIV DRMs. HIV-1 drug resistance distribution was homogeneous across the three study regions.

Authors’ response: Thanks

Other findings are less statistically sound and need to be corrected or repealed; for example, the phrases “This abovementioned table also indicated the predominance of pure HIV-1 subtype  (71.4%, of which A = 50.0%, C = 30.0%, D = 10.0%, G = 10.0%) compared with HIV-1 recombinant forms in FSWs that had a high HIV-1 VL, while this HIV-1 strain was equally 209 distributed (50.0%) in FSWs that had a low HIV-1 VL score” and “HIV-1 recombinant forms were less affected by 288 NRTIs” clearly need additional statistical support.

Authors’ response: Thanks for this valuable comment. We have deleted this statement.

The main limitation of the work, as noted by the authors, is the lack of information about the treatment of patients. This makes it impossible to draw conclusions about the prevalence of mutations before treatment start. In this regard, the statement "around one-fifth of FSWs could fail to respond to first-line ART due to HIVDRMs" needs to be corrected. It would be more correct to say that one-fifth of FSWs have a virus that is resistant to treatment with drugs of first-line ART.

Authors’ response: Thanks for the comment. We have edited accordingly.

There are a few typos in the text.

Authors’ response: Thanks for the comment. We have edited accordingly

The article, in our opinion, needs correction and subsequent publication.

Authors’ response: Thanks for the comment. We made what all reviewers requested as edits.

Round 2

Reviewer 1 Report

My concerns are well addressed.